# Injury-induced pulmonary tuft cells are heterogenous, arise independent of key Type 2 cytokines, and are dispensable for dysplastic repair

Justinn Barr[1†], Maria Elena Gentile[2,3,4†], Sunyoung Lee[1], Maya E Kotas[5], Maria Fernanda de Mello Costa[2], Nicolas P Holcomb[2], Abigail Jaquish[1], Gargi Palashikar[2], Marcella Soewignjo[2], Margaret McDaniel[6], Ichiro Matsumoto[7], Robert Margolskee[7], Jakob Von Moltke[6], Noam A Cohen[7,8,9], Xin Sun[10]*, Andrew E Vaughan[2,3,4]*

[1]Department of Pediatrics, University of California, San Diego, San Diego, United States; [2]Department of Biomedical Sciences, School of Veterinary Medicine, University of Pennsylvania, Philadelphia, United States; [3]Institute for Regenerative Medicine, University of Pennsylvania, Philadelphia, United States; [4]Lung Biology Institute, University of Pennsylvania, Philadelphia, United States; [5]Division of Pulmonary, Critical Care, Allergy & Sleep Medicine, University of California, San Francisco, San Francisco, United States; [6]Department of Immunology, University of Washington, Seattle, United States; [7]Monell Chemical Senses Center, Philadelphia, United States; [8]Department of Otorhinolaryngology-Head and Neck Surgery, University of Pennsylvania, Perelman School of Medicine, Philadelphia, United States; [9]Corporal Michael J. Crescenz Veterans Administration Medical Center Surgical Service, Philadelphia, United States; [10]Division of Biological Sciences, University of California, San Diego, San Diego, United States

*For correspondence:
xinsun@health.ucsd.edu (XS);
andrewva@vet.upenn.edu (AEV)

†These authors contributed equally to this work

**Abstract** While the lung bears significant regenerative capacity, severe viral pneumonia can chronically impair lung function by triggering dysplastic remodeling. The connection between these enduring changes and chronic disease remains poorly understood. We recently described the emergence of tuft cells within Krt5+ dysplastic regions after influenza injury. Using bulk and single-cell transcriptomics, we characterized and delineated multiple distinct tuft cell populations that arise following influenza clearance. Distinct from intestinal tuft cells which rely on Type 2 immune signals for their expansion, neither IL-25 nor IL-4ra signaling are required to drive tuft cell development in dysplastic/injured lungs. In addition, tuft cell expansion occurred independently of type I or type III interferon signaling. Furthermore, tuft cells were also observed upon bleomycin injury, suggesting that their development may be a general response to severe lung injury. While intestinal tuft cells promote growth and differentiation of surrounding epithelial cells, in the lungs of tuft cell deficient mice, Krt5+ dysplasia still occurs, goblet cell production is unchanged, and there remains no appreciable contribution of Krt5+ cells into more regionally appropriate alveolar Type 2 cells. Together, these findings highlight unexpected differences in signals necessary for murine lung tuft cell amplification and establish a framework for future elucidation of tuft cell functions in pulmonary health and disease.

## Editor's evaluation

In this manuscript, Barr and colleagues report some novel and surprising results in regards to the development and role of tuft cells during influenza-induced lung injury. The authors demonstrate how unlike in the intestine lung tuft cells do not require Il-25, Il-4Ra, or Trmp5 but do require Pou2f3. Interestingly, loss of tuft cells in Pou2f3 null mice did not affect basal cell or goblet cell differentiation in basal cell pods, suggesting that additional studies are required to better understand the functional significance of these interesting cells.

## Introduction

The lung exhibits a remarkable capacity for repair following damage induced by either pathogen infection (e.g., influenza [*Kumar et al., 2011*], SARS-CoV-2 [*Fang et al., 2020*]) or sterile injury (e.g., pneumonectomy [*Ding et al., 2011*], bleomycin [*Cong et al., 2020*]). Within the gas-exchanging alveoli, normally quiescent tissue-resident alveolar Type 2 cells (AT2s) can self-renew and differentiate into alveolar Type 1 cells (AT1s) upon mild injury, providing a source of oxygen-exchanging epithelium for effective repair (*Barkauskas et al., 2013*; *Evans et al., 1975*). However, upon severe lung injury, for example, that caused by H1N1 influenza infection, large regions of alveolar epithelium can be ablated. In its place, dysplastic tissue arises composed of cytokeratin 5 (Krt5)$^+$ p63$^+$ 'basal-like' cells, forming 'epithelial scars' that appear to provide a short-term benefit in restoring barrier function. However, these cells rarely differentiate into AT2s or AT1s capable of gas exchange (*Fernanda de Mello Costa et al., 2020*; *Vaughan et al., 2015*; *Xi et al., 2017*; *Zuo et al., 2015*), so dysplastic repair processes may prioritize rapid barrier restoration at the expense of proper lung function. While pathologic changes and diminished lung function induced by influenza infection can persist long after viral clearance, the mechanistic basis for chronic post-viral disease remains unclear.

Tuft cells, which depending on their anatomic location, are also known as brush cells (trachea), microvillus cells (olfactory epithelium), or solitary chemosensory cells (sinonasal respiratory epithelium), are rare cells at homeostasis and were discovered over 50 years ago by electron microscopy based on their unique morphology in the rodent gastrointestinal tract and airway (*Billipp et al., 2021*; *Jarvi and Keyrilainen, 1956*; *Rhodin and Dalhamn, 1956*). Tuft cells are non-ciliated epithelial cells that exhibit a bottle-shaped morphology with apical microvilli that extend into the lumen of mucosal organs (*Rhodin and Dalhamn, 1956*; *Schneider et al., 2019*). Early reports relied entirely on their unique morphology to distinguish them in different tissues/organs, without an understanding of their function. Expression profiling of murine intestinal tuft cells using a transient receptor potential cation channel subfamily M member 5 (Trpm5)-GFP reporter (*Bezençon et al., 2008*) suggested that tuft cells have a role in chemosensory, immune, and neuronal pathways, the latter two of which are not typically associated with epithelial cells. In major paradigm-building work, tuft cells were recently identified in the gastrointestinal tract as being important for initiating Type 2 immunity and epithelial tissue remodeling (*Gerbe et al., 2016*; *Howitt et al., 2016*; *von Moltke et al., 2016*). In addition, tuft cells were found to be the sole producers of interleukin (IL)-25, needed to activate the ILC2-circuit required for promoting anti-parasitic immune responses (*Gerbe et al., 2016*; *Howitt et al., 2016*; *von Moltke et al., 2016*). It was also determined that tuft cell expansion in the gastrointestinal tract requires this Type 2 immune response, specifically IL-25 and IL-4ra signaling (*Gerbe et al., 2016*; *Howitt et al., 2016*; *von Moltke et al., 2016*). Additionally, tuft cell specification depends on the master transcription factor POU domain, class 2, transcription factor 3 (Pou2F3) (*Gerbe et al., 2016*; *Ohmoto et al., 2013*; *Yamaguchi et al., 2014*; *Yamashita et al., 2017*), also required for the development of type II/bitter taste bud cells (*Matsumoto et al., 2011*), to which tuft cells are very closely related.

Our group recently described the ectopic development of tuft cells in H1N1 influenza A virus (IAV; PR8 strain)-injured murine lungs, which are normally present only in the central airways and absent from distal portions of healthy lungs (*Rane et al., 2019*). Tuft cells were identified specifically within dysplastic Krt5$^+$ epithelial regions along the airway and in injured alveoli (*Rane et al., 2019*), since corroborated by additional reports (*Roach et al., 2022*), suggesting that they may contribute to the development or persistence of inappropriately remodeled regions of tissue and thus participate in chronic pulmonary dysfunction. Importantly, tuft cell expansion has recently been demonstrated to occur after severe SARS-CoV-2 infection in humans as well (*Melms et al., 2021*), possibly contributing to 'long COVID' pulmonary symptoms, a significant and growing public health concern. In addition, the presence of tuft cells in dysplastic regions of the lung correlated with features of a chronic Type

2 immune response, such as increased eosinophilia, goblet cell hyperplasia, and IL-13 long after viral clearance (*Keeler et al., 2018*; *Rane et al., 2019*), raising potential concern for their possible contribution to post-viral reactive airway disease and/or chronic inflammation.

In this study we aimed to transcriptionally characterize ectopic tuft cells that develop as a result of influenza-induced lung epithelial remodeling, and to determine if their emergence requires the same Type 2 cytokine signals as those observed in the small intestine. Using bulk and single-cell RNA sequencing (RNA-Seq), we identified transcriptionally heterogenous tuft cells that arise following IAV infection. In contrast to intestinal tuft cells, post-IAV lung tuft cells arise independent of IL-25 or IL-4ra signaling. In addition, *Pou2f3*[-/-] mice, which are unable to develop tuft cells post-IAV infection, still exhibit dysplastic (Krt5[+]) epithelial remodeling, and we did not observe any increase in conversion to AT2 cells in those regions. Finally, we demonstrate that tuft cells also arise in Krt5[+] areas upon bleomycin injury, albeit more sparsely, suggesting their emergence may be a general feature of severe lung injury.

## Results

### Transcriptional profiling of post-influenza pulmonary tuft cells

We recently demonstrated the establishment of ectopic tuft cells in severely injured airways and alveolar areas post-IAV injury, which appear in locally large concentrations within dysplastic/remodeled Krt5[+] regions (*Figure 1A*; *Rane et al., 2019*). To further investigate the nature of these cells, we utilized Trpm5-GFP reporter mice and sorted CD45[low/neg] EpCAM[pos] GFP[pos] cells from post-IAV lungs followed by either 'bulk' RNA-Seq on the total population or single-cell RNA-Seq to reveal potential tuft cell heterogeneity (*Figure 1B–C*). Bulk RNA-Seq (*Figure 1D–E*) revealed a transcriptomic signature highly conserved with both intestinal and tracheal tuft cells (*Nadjsombati et al., 2018*), including enrichment in key tuft cell markers including *Trpm5*, *Sox9*, and *Dclk1* (*Figure 1D–E*). In total, 898 genes were differentially expressed between purified tuft cells and the remaining lung epithelium (adjusted p-value <0.05). To directly compare influenza-induced lung tuft cells with tracheal tuft cells, we compared our bulk RNA-Seq data with published bulk RNA-Seq data from sorted tracheal tuft cells (*Figure 1—figure supplement 1*; *Nadjsombati et al., 2018*). Canonical tuft cell genes are expressed by post-influenza lung tuft cells at similar levels to tracheal tuft cells. An apparent distinction is reduced expression of *Chat* and *Plcb2* by post-influenza lung tuft cells (*Figure 1—figure supplement 1*).

To evaluate tuft cell heterogeneity, we analyzed our single-cell RNA-Seq data on these cells (*Figure 2A–D*). To further enrich for tuft cells, we restricted analysis to cells expressing detectable *Trpm5*. Among these, we observed three to five populations of cells depending upon clustering variables (*Figure 2A*). One of these populations (cluster 2) co-expressed basal cell markers Krt5 and Trp63, suggesting that these were basal cells very early in their differentiation toward tuft cells, in agreement with our earlier demonstration that all post-IAV tuft cells are derived from intrapulmonary p63[+] basal-like cells (*Rane et al., 2019*). To further confirm that tuft cells are derived from basal-like cells after flu, we performed lineage tracing using Krt5-CreERT2 and found that 90% of Dclk1[+] cells were labeled in the alveoli, with tdTomato signal clearly visible in tuft cell nuclei (*Figure 2E*, n=323 Dclk1[+] cells). Another population (merged clusters 0, 3, 4) bore relatively high mitochondrial gene reads, which we interpret as 'stressed' cells that nonetheless passed the quality control thresholds of data processing in Seurat. Whether this population is biologically relevant or a technical artifact is difficult to assess, though this 'stressed' population does contain subpopulations enriched for selected genes (*Figure 2A, B*).

The remaining population (largely cluster 1) bore neither stress-related genes nor basal cell genes and appeared to be heterogenous based on distribution in UMAP space (*Figure 2A*). Recent comprehensive single-cell analysis of the naïve murine tracheal airway epithelium suggested the existence of two distinct tuft cell subtypes denoted 'Tuft-1' (enriched in gustatory pathway genes) and 'Tuft-2' (enriched in leukotriene synthesis genes) (*Montoro et al., 2018*). Utilizing published gene sets for these two tuft cell subtypes, we performed gene module enrichment in Seurat, revealing a small subpopulation of 'Tuft-1' cells uniquely expressing known Tuft-1 genes, and a larger subpopulation of 'Tuft-2' cells, whose gene expression is relatively broad, but is expressed higher in this subpopulation than in 'Tuft-1' subpopulation (*Figure 2D*, *Figure 2—figure supplement 1*). To corroborate these as distinct identities in vivo, we performed immunostaining for 'Tuft-1' marker Gnb3 in Trpm5-GFP

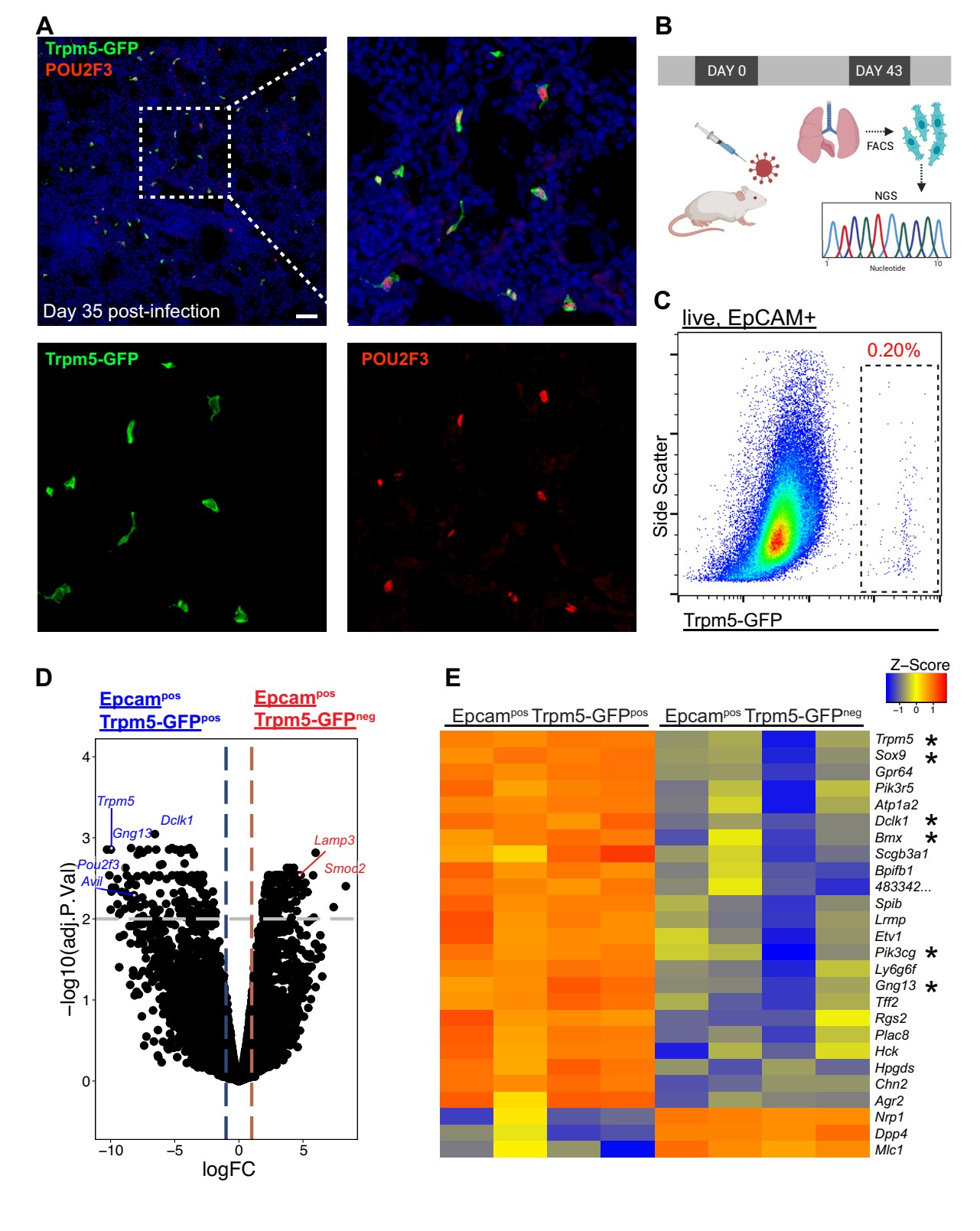

**Figure 1.** Epcam+ Trpm5-GFP+ cells are bona fide tuft cells in the lung post-influenza. (**A**) Representative immunostaining of lung sections from Trpm5-GFP reporter mice at day 35 post-influenza. Nuclear stain (DAPI) in blue, Trpm5-GFP in green, and POU2F3 in red. (**B**) Experimental design outlining the bulk RNA sequencing (RNA-Seq) experiment. (**C**) Trpm5-GFP reporter expression in live lung epithelial (Epcam+) cells post-influenza via FACS. (**D**)

*Figure 1 continued on next page*

*Figure 1 continued*

Volcano plot and (**E**) heatmap comparing gene expression between Epcam$^+$Trpm5-GFP$^+$ (tuft cells) and Epcam$^+$Trpm5-GFP$^-$ (non-tuft epithelial) cells from mice at day 43 post influenza.*=Representative genes that have been previously associated with tuft cells.

The online version of this article includes the following figure supplement(s) for figure 1:

**Figure supplement 1.** Gene expression comparison of post-influenza A virus (IAV) lung tuft cells and tracheal cells (*Nadjsombati et al., 2018*) using bulk RNA sequencing data.

reporter mice, observing distinct Gnb3 expression in approximately 20% of Trpm5-GFP$^+$ cells in situ (*Figure 2F*), though we did not observe any preferential localization of Gnb3$^+$ cells.

To further investigate the influenza-induced lung tuft cell lineage, we utilized Slingshot to identify trajectories (*Figure 2—figure supplement 2*). Consistent with lineage tracing results, a trajectory from the 'basal ->tuft' cluster to the 'Tuft-2' and 'Tuft-1' clusters was identified, with the trajectory ending on the 'stressed' cluster (*Figure 2—figure supplement 2B–C*). A second trajectory analysis program, Monocle, produced similar pseudotime results (*Figure 2—figure supplement 2D*). Taken together, we conclude that injury-induced ectopic lung tuft cells are, like their homeostatic counterparts in the trachea, transcriptionally heterogenous, though the biological relevance of this heterogeneity remains to be determined.

## Post-injury tuft cells are dependent on Pou2f3 but arise independently of key intestinal 'tuft cell circuit' cytokines IL-25 and IL4/IL-13, and are dispensable for the generation of dysplastic Krt5$^+$ cells

In the small intestine, tuft cells are present in small numbers during homeostasis, but their prevalence increases rapidly upon infection with various Th2-associated pathogens in a manner dependent upon ILC2-derived IL-13 and tuft cell-derived IL-25. These cytokines facilitate a feed-forward loop to promote increased differentiation of Lrg5$^+$ stem cells into tuft cells, ultimately increasing tuft cell numbers alongside increased fractions of goblet cells to promote pathogen clearance (*Gerbe et al., 2016*; *Howitt et al., 2016*; *von Moltke et al., 2016*). We therefore performed IAV infection in *Pou2f3$^{-/-}$*, *Trpm5$^{-/-}$* (required for tuft cell chemosensing), *Il4ra$^{-/-}$* (co-receptor required for both IL-13 and IL-4 signaling) and *Il25$^{-/-}$* knockout animals. As expected, *Pou2f3$^{-/-}$* mice entirely failed to develop lung tuft cells (*Figure 3A–C*, *Figure 3—figure supplement 1*). In *Trpm5$^{-/-}$* mice, tuft cells still differentiate after IAV injury, and we did not observe a significant change in tuft cell number at 25 days post injury (*Figure 3E–G*, *Figure 3—figure supplement 2A, C*, *Figure 3—figure supplement 3B*). Though we anticipated a recapitulation of the circuit found in the small intestine, we observed no difference in total numbers of tuft cells in either *Il4ra$^{-/-}$* (*Figure 3I–K*, *Figure 3—figure supplement 2B, D*, *Figure 3—figure supplement 3C*) or *Il25$^{-/-}$* animals (*Figure 3M–O*). An alternative strategy to eliminate IL-4ra signaling, using an inducible, ubiquitously expressed Cre and floxed *Il4ra* allele, again resulted in no significant difference in tuft cell numbers compared with controls (*Figure 3—figure supplement 4*). Consistent with independence from Type 2 cytokines, in normal mice following infection, we did not observe significantly increased levels of *Il4*, *Il5*, or *Il13*, at timepoints when tuft cells are present in the lung – 12, 21, or 51 days post infection, although we did observe significant induction of Type 2 cytokines at earlier timepoints (*Figure 3—figure supplement 5A*). Additionally, we found that the loss of tuft cells in the *Pou2f3$^{-/-}$* mice had no effect on viral clearance (*Figure 3—figure supplement 5B*), in line with the timing that flu-induced tuft cells arise at timepoints well after viral clearance (*Rane et al., 2019*).

To address if the presence or absence of post-injury tuft cells in the various mutants altered the formation of Krt5$^+$ dysplastic cells, we systematically assayed the percentage of Krt5$^+$ cell area in total lung area by scanning and quantifying whole lobe sections to ensure representation of all regions of the lung (*Figure 3—figure supplements 1–3*). We observed the expected variation of the percentages due to variable response to infection from animal to animal in all the mutants and corresponding controls. In *Trpm5$^{-/-}$*, *Il4ra$^{-/-}$*, and *Il25$^{-/-}$* mutants where tuft cells are present, we noted no apparent change in overall Krt5$^+$ area of the injured lung compared to corresponding controls (*Figure 3H, L and P*). There was also no statistically significant difference of overall Krt5$^+$ area in *Pou2f3$^{-/-}$* mutant where tuft cells are not present compared to control (*Figure 3D*). Consistently, we did not detect a significant difference in the percent of proliferating (Ki67$^+$) alveolar Krt5$^+$ basal-like cells at 14 or 25 days post

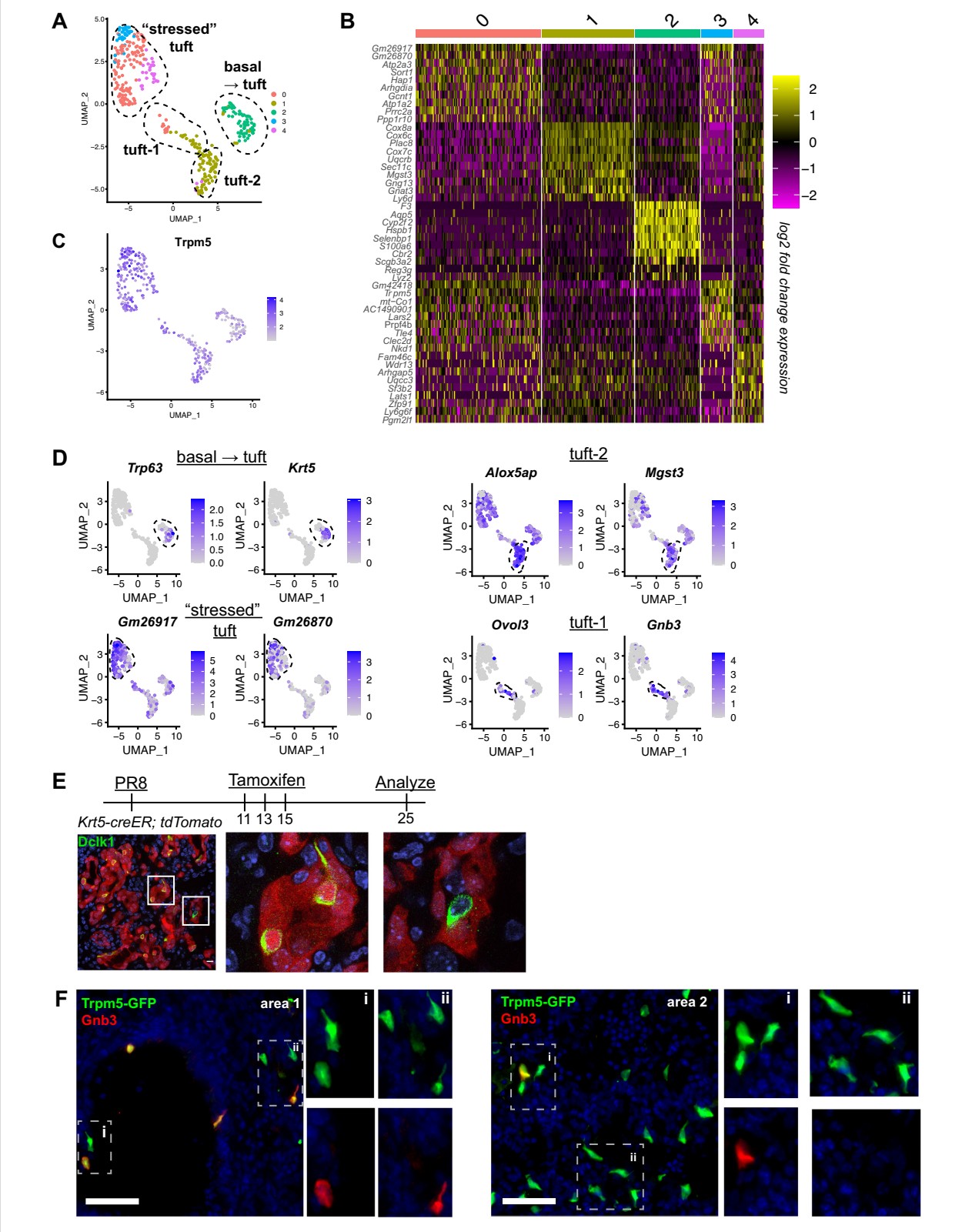

**Figure 2.** Distinct tuft cell populations derived from Krt5+ cells arise in the lung following influenza clearance. (**A**) Single-cell RNA-Seq UMAP clustering of sorted tuft cells (Epcam+Trpm5-GFP+) from Trpm5-GFP reporter mice at day 28 post influenza. (**B**) Heatmap comparing the gene expression profile between the tuft cell clusters identified in A. (**C**) Trpm5 expression highlighted in all analyzed cells, confirming all analyzed cells are tuft cells and not contaminating cells. (**D**) Marker gene expression highlighted within the different tuft cell UMAP clusters. (**E**) Representative immunostaining of

*Figure 2 continued on next page*

*Figure 2 continued*

tamoxifen-treated Krt5-CreER tdTomato lungs 25 days post influenza. Nuclear stain (DAPI) in blue, DCKL1 in green, Krt5-CreER tdTomato lineage label in red. Lineage traced tuft cells (tdTomato+DCKL1+) appear yellow. (**F**) Representative immunostaining for Gnb3 (Tuft-1 signature) in the Trpm5-GFP reporter mice 35 days post influenza. Nuclear stain in blue, Gnb3+ cells in red, Trpm5-GFP+ cells in green, and double positive cells (Gnb3+ and Trpm5+) in yellow. Single color insets shown (**i, ii**).

The online version of this article includes the following figure supplement(s) for figure 2:

**Figure supplement 1.** Further analysis of the different tuft cell clusters utilizing previously identified transcriptomic signatures.

**Figure supplement 2.** Trajectory analysis of influenza A virus (IAV)-induced lung tuft cells from single-cell data.

infection (*Figure 3—figure supplement 5C-D*). While we cannot rule out differences in the behavior of other reparative cells, these findings indicate that post-injury tuft cells and Th2 immune signals (at least IL-4, IL-13, and IL-25) are apparently dispensable for the formation of Krt5+ cells in injured lungs. We note that experiments utilizing *Pou2f3-/-* and *Il4ra-/-* mice were performed independently in two separate vivariums at different institutions with equivalent outcomes, indicating stability of the phenotype with no obvious impact of potentially distinct housing environments (*Figure 3—figure supplement 3A*).

## Type I (α/β) and type III (λ) interferon signaling are dispensable for the development of post-injury tuft cells

Type I (α/β) and type III (λ) interferons are highly expressed following influenza infection (*Killip et al., 2015*) and are crucial components of the innate anti-viral response (*Pardy et al., 2019*). As interferons play a crucial role early in flu infection and the action of type III interferon is restricted to the epithelium (*Killip et al., 2015*), we assessed whether they play a role in tuft cell development post IAV infection. Since IFNAR-deficient mice have been shown to have increased mortality when infected with flu (*Arimori et al., 2013*; *Killip et al., 2015*; *Seo et al., 2011*), we infected *Ifnar1*- and *Il28r*-deficient mice with a lower dose of PR8 than their BL/6 controls. Using this lower dose, the interferon receptor-deficient mice had an average weight loss of 22%, comparable morbidity to their BL/6 counterparts (*Figure 4—figure supplement 1*), and still bore abundant Krt5+ patches. In the absence of type III or type I interferon signaling, we observed no significant difference in tuft cell numbers following flu infection compared to their BL/6 comparators at 22 days post infection (*Figure 4A-F*). Therefore, type I and type III interferon signaling are dispensable in the development of tuft cells following PR8 infection.

## Bleomycin injury also facilitates ectopic tuft cell development

Following the results that IFN signaling, which is commonly linked to viral infection, is not required for tuft cell formation, we sought to determine whether ectopic tuft cell development in the lung was specific to viral infection. To test if this is a general phenomenon occurring following lung injury, we treated mice with bleomycin, a chemotherapeutic agent that induces lung damage and subsequent Krt5+ areas in the lung (*Vaughan et al., 2015*). We identified tuft cells (Dclk1+) within Krt5+ patches in the lungs of bleomycin-treated mice by immunostaining, albeit sparser than in IAV-infected mice (*Figure 4G*, n=3 mice). These data demonstrated that tuft cell development can occur independently of infection and is a result of lung injury.

## Tuft cells do not influence goblet cell differentiation after influenza infection

Amplification of tuft cells in the intestine promotes goblet cell metaplasia through Th2 cytokines (*Gerbe et al., 2016*; *von Moltke et al., 2016*). To address if the ectopic lung tuft cells and cytokines may perform a similar role, we stained for Agr2 (anterior gradient 2) as a cellular marker of goblet cells at 25 days after infection, a stage when goblet cells are robustly present (*Chen et al., 2009*; *Di Valentin et al., 2009*; *Rane et al., 2019*). To minimize variation due to the extent of injury, we normalized the area of Agr2+ cells to the area of Krt5+ cells. We found that in *Pou2f3-/-*, *Trpm5-/-*, and *Il4ra-/-* animals, the percent of Agr2+ area within Krt5+ area was not significantly different from controls following influenza infection (*Figure 5A–I*). Consistent with this, Muc5b immunostaining in damaged regions of lung did not differ between *Pou2f3-/-* or *Il4ra-/-* lungs 25 days post infection (*Figure 5J–M*). In

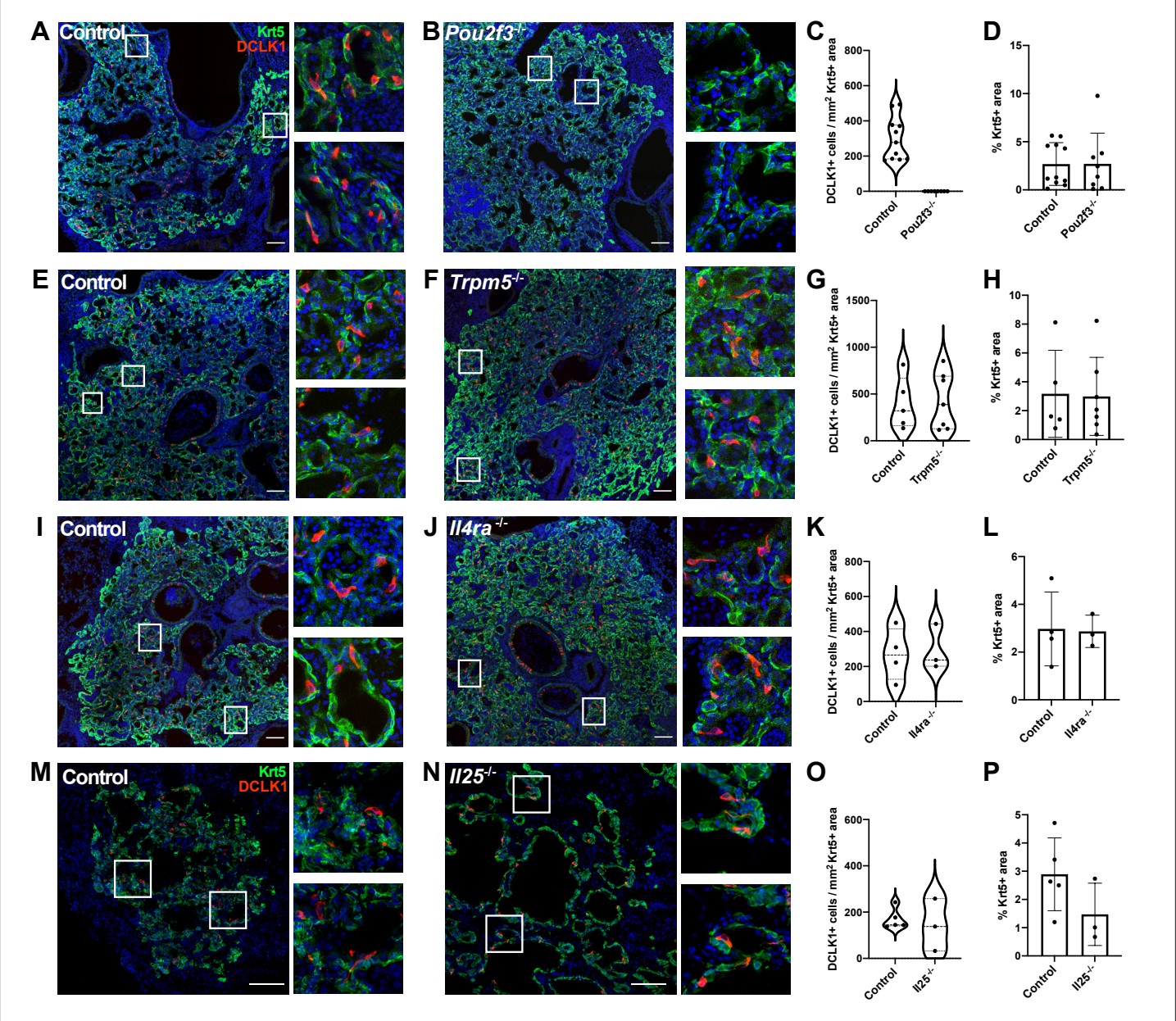

**Figure 3.** Tuft cells are not required for the epithelial dysplastic response after lung injury. (**A–P**) Lung sections stained for Krt5 in green, Dclk1 in red, and DAPI in blue 22–25 days after influenza. (**A–D**) Dclk1+ cells are absent in *Pou2f3*-/- (n=8) compared to control (n=11), without a significant change in Krt5+ area. (**E–P**) No significant difference was found in the number of Dclk1+ cells per Krt5+ area or percent Krt5+ lung area when comparing (**E–H**) control (n=5) and *Trpm5*-/- (n=7), (**I–L**) control (n=4) and *Il4ra*-/- (n=3), or (**M–P**) wild type (WT) (n=5) and *Il25*-/- (n=3). Dclk1+ cells per Krt5+ area was quantified in (**C, G, K, O**) and was not statistically significantly different in (**G, K, O**).Changes in percent Krt5+ area were also not statistically significant in (**D, H, L**, P). (**A–L**) Analysis 25 dpi, (**M–P**) analysis 22 dpi. (**A–B, E–F, I–J, M–N**) Scale bar is 100 µm, images are cropped from a multipanel stitched image.

The online version of this article includes the following source data and figure supplement(s) for figure 3:

**Source data 1.** Tuft cells per Krt5 area and percent Krt5 area in controls and *Pou2f3*-/-, *Trpm5*-/-, *Il4ra*-/-, and *Il25*-/-.

**Figure supplement 1.** Examples of *Pou2f3* control and mutant Krt5+ regions.

**Figure supplement 1—source data 1.** Percent body weight loss of control and *Pou2f3*-/- mice 7, 9, and 16 days post influenza infection.

**Figure supplement 2.** Examples of *Trpm5* and *Il4ra* control and mutant Krt5+ regions.

**Figure supplement 3.** Analysis of tuft cell density and Krt5+ area.

**Figure supplement 3—source data 1.** Tuft cells per Krt5 area and percent Krt5 area.

**Figure supplement 4.** Tuft cell differentiation does not depend on IL-4ra signaling.

*Figure 3 continued on next page*

*Figure 3 continued*

**Figure supplement 4—source data 1.** Analysis of *Il4ra* conditional knockout mice.

**Figure supplement 5.** Tuft cell differentiation does not impact basal-like cell proliferation.

**Figure supplement 5—source data 1.** Time course of Type 2 cytokine expression, lung viral titer, and analysis of basal-like cell proliferation.

order to quantify mucus metaplasia in whole lungs of control and *Pou2f3⁻/⁻* animals, we analyzed gene expression of goblet cell markers, *Foxa3, Agr2, Muc5ac,* and *Muc5b*, 51 days post infection. We did not observe a significant difference in relative expression of goblet cell markers when normalizing to *Actb* or to *Krt5* as a marker of tissue damage (*Figure 5N*).

## Tuft cells do not overtly impact alveolar differentiation/plasticity of dysplastic Krt5⁺ epithelial remodeling

After IAV injury, basal-like cells are rarely observed to act as progenitors for AT1s or AT2s, though they do so at a higher frequency after bleomycin injury as indicated by lineage tracing with Krt5-creERT2 labeling AT2s (*Vaughan et al., 2015*; *Yuan et al., 2019*). As tuft cell differentiation is more heterogeneous in bleomycin compared with influenza, we investigated whether tuft cells are involved in preventing normal alveolar differentiation in Krt5⁺ areas. As shown above, we observed no change in total Krt5⁺ area of the of lung with Pou2f3 deletion (*Figure 3A–D*). To address if a relatively minor population of Krt5⁺ cells may be converting to AT2s, we double stained for Krt5 and the AT2 cell marker SPC⁺ in *Pou2f3⁻/⁻* and control lungs at 25 days after infection to label cells in the process of conversion (*Figure 6A*). We did not observe any overlap of staining in either *Pou2f3⁻/⁻* or control lungs. To more rigorously trace possible conversion, we bred *Pou2f3⁻/⁻* mice to Krt5-CreERT2 fate mapping mice to lineage trace Krt5⁺ cells and assess whether any increased plasticity may occur in the absence of tuft cells (*Figure 6B*). Unlike tuft cells, which could be lineage traced from Krt5⁺ precursors, we did not observe any appreciable conversion of Krt5⁺ precursors into AT2 (SPC⁺) cells in the presence or absence of tuft cells (*Figure 6C–D*). Taken together, while ectopic lung tuft cells likely possess as-of-yet undiscovered function in post-IAV lungs, it does not appear that they play an important role in restricting Krt5⁺ epithelial cell plasticity, nor are they responsible for the inability of Krt5⁺ cells to efficiently differentiate into more regionally appropriate alveolar cell types.

## Discussion

Epithelial tuft cells are sensory cells that can modulate neural and immune responses. In this study, we determined the transcriptional profiles of the influenza-induced lung tuft cells that we have identified previously (*Rane et al., 2019*). Our single-cell RNA-Seq data revealed heterogeneity among these lung tuft cells, including a 'basal/tuft' hybrid cluster, consistent with lineage tracing Dclk1⁺ tuft cells from p63⁺/Krt5⁺ basal progenitors, and 'Tuft-1' and 'Tuft-2' clusters, distinct subtypes enriched for genes related to sensory functions and eicosanoid synthesis, respectively. Additionally, we observed a cluster of tuft cells which appeared enriched in 'stress' markers, though we remain agnostic as to the in vivo relevance of these cells or whether they arise from the stress associated with enzymatic dissociation and sorting during the single-cell analysis procedure.

Similar to tuft cells in the intestine, we found that the formation of lung tuft cells depends on the cell type-defining transcription factor gene *Pou2f3*. However, despite this shared dependence and common gene expression signatures, we identified major differences in the behavior of lung tuft cells compared to their intestinal counterpart. Unlike intestinal tuft cells which are dependent on Th2 cytokines for their amplification, lung tuft cells arise in similar numbers in *Il25⁻/⁻* or *Il4ra⁻/⁻* mutants compared to controls following infection. Moreover, while intestinal tuft cells strongly influence the growth and differentiation of surrounding epithelial cells in infectious/inflammatory settings, lung tuft cells appear dispensable for the formation of Krt5⁺ cells from p63⁺ basal-like cells, their migration into damaged alveoli, and goblet cell metaplasia. We arrive at these results through a combination of manual and automated quantification to account for both the irregular tuft cell shape and patchiness of infection (*Figure 3*, *Figure 3—figure supplement 3*). The interpretations of experimental outcomes in *Pou2f3⁻/⁻* and *Il4ra⁻/⁻* mice are further strengthened by the fact that our two groups performed these experiments entirely independently in separate vivariums, yet observed the same conclusions, arguing

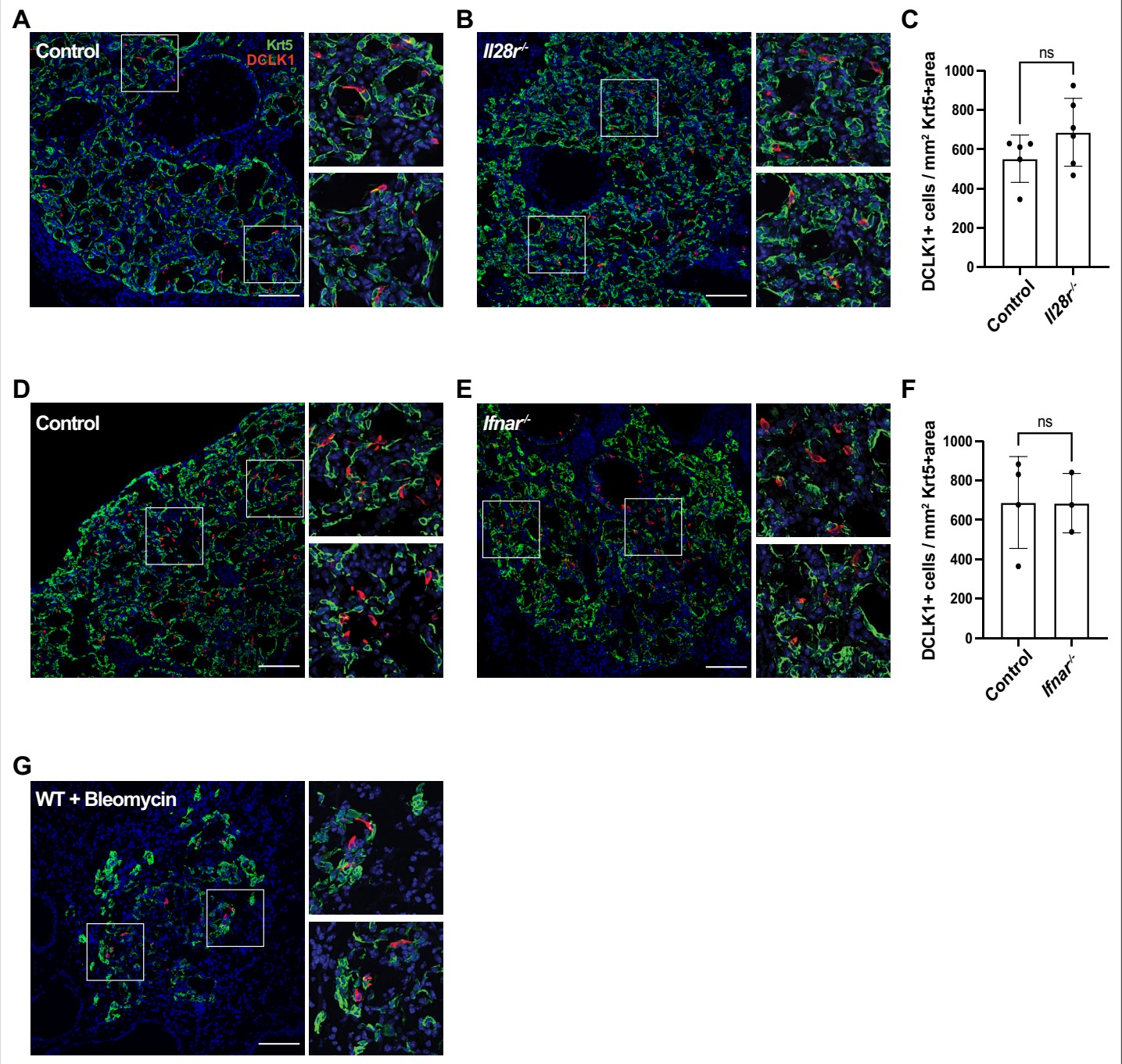

**Figure 4.** Type I and type III interferon signaling are dispensable for tuft cell development after influenza infection. (**A–B, D–E**) Lung sections stained for Krt5 in green, Dclk1 in red, DAPI in blue 22 days after influenza infection. (**A–F**) No significant differences were found in the Dclk1⁺ cells number per Krt5⁺ area when comparing (**A–C**) wild type (WT) (n=5) and *Il28r⁻/⁻* (n=6) mice and (**D–F**) WT (n=4) and *Ifnar⁻/⁻* (n=3). (**A–C**) and (**D–F**) are each pooled from two independent experiments. (**G**) Representative image of a lung section from a WT mouse (n=3) treated with bleomycin stained for Dclk1 in red, Krt5 in green, and DAPI in blue, 22 days following injury. (**A–B, D–E, G**) Scale bar is 100 µm, images are cropped from the 20x z-stack image. Error bars represent standard deviation.

The online version of this article includes the following source data and figure supplement(s) for figure 4:

**Source data 1.** Tuft cells per Krt5 area in controls and *Il28r⁻/⁻* and *Ifnar1⁻/⁻*.

**Figure supplement 1.** Interferon receptor-deficient mice have comparable weight loss to BL/6 mice when infected with a lower PR8 dose.

**Figure supplement 1—source data 1.** Weight loss post influenza A virus (IAV) infection of controls and *Il28r⁻/⁻* and *Ifnar1⁻/⁻*.

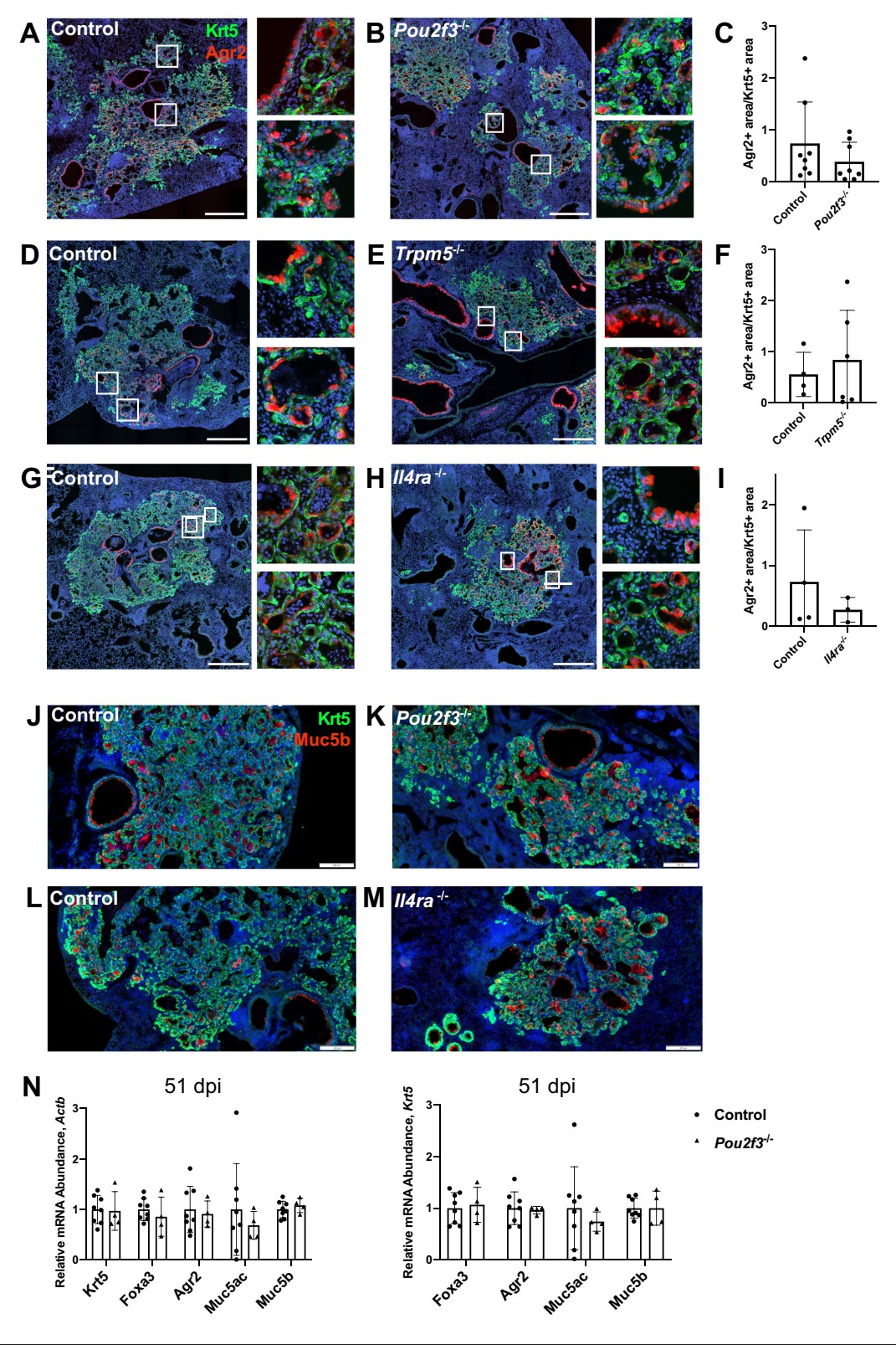

**Figure 5.** Tuft cells are not required for goblet cell differentiation after influenza. (**A–I**) Krt5 (green) and Agr2 (red) staining and quantification 25 days after influenza. Agr2$^+$ area per Krt5$^+$ area was not significantly different between lung sections of (**C**) control (n=8) and *Pou2f3$^{-/-}$* (n=8), (**F**) control (n=4) and *Trpm5$^{-/-}$* (n=6) and (**I**) control (n=3) and *Il4ra$^{-/-}$* (n=3). (**J–M**) Krt5 (green) and Muc5b (red) staining 25 days after influenza demonstrates Muc5b staining

*Figure 5 continued on next page*

*Figure 5 continued*

in (**J–K**) control (n=4) and *Pou2f3*$^{-/-}$ (n=3) and (**L–M**) control (n=4) and *Il4ra*$^{-/-}$ (n=3) dysplastic alveolar regions and (**N**) qRT-PCR (quantitative RT-PCR) for relative mRNA levels for goblet cell markers in control (n=8) and *Pou2f3*$^{-/-}$ (n=4) lungs 51 days after influenza, expression normalized to *Actb* (left) and *Krt5* (right). (**A–H**) Scale bar is 500 µm, (**J–M**) scale bar is 200 µm.

The online version of this article includes the following source data for figure 5:

**Source data 1.** Quantification of Agr2 immunostaining and expression of goblet cell transcripts.

against a prominent role for differing housing environments in directing development and function of ectopic tuft cells.

In both the small intestine and the trachea, tuft cell production of immune signaling ligands including cytokines such as IL-25 and leukotriene such as LTC$_4$ play important roles for initiating a positive feedback circuit resulting in increased tuft cell differentiation. In the trachea, i.p. administration of IL-13 was found to increase tuft cell numbers (*Ualiyeva et al., 2021*), and baseline tuft cell numbers were reduced in *Stat6*$^{-/-}$ mice, with blunted response to Th2 signals (*Bankova et al., 2018*). Moreover, IL-13 promotes prostaglandin E2 generation in upper airway tuft cells, in turn promoting CFTR-mediated mucocilliary transport (*Kotas et al., 2022*). In contrast to these findings which suggest that IL-13 signaling influences upper airway tuft cell differentiation, our data in distal lung indicate that IL-13 plays no role in the development of tuft cells after injury and remodeling, a somewhat surprising discrepancy given the cellular similarity between the dysplastic, 'bronchiolized' Krt5$^+$ regions and tracheal epithelium. Other aspects of lung remodeling after IAV infection, including airway hyperreactivity and airway mucus production, are partially attenuated in *Il13* knockout mice (*Keeler et al., 2018*). However, mucus is not significantly reduced within the most severely damaged regions of the distal lung (*Keeler et al., 2018*). Our results build upon these findings and cumulatively indicate that while some outcomes of pathologic lung remodeling after IAV infection are dependent on Type 2 cytokines, others are independent.

In the trachea, tuft cell differentiation following challenge with the allergen *Alternaria* is dependent on leukotriene signaling, and leukotriene LTE$_4$ administration is sufficient to increase tuft cells (*Bankova et al., 2018*). This increase is independent of *Stat6*, suggestive of a distinct pathway from that of the Th2 cytokines (*Bankova et al., 2018*). In the lung, whether IAV infection acts through leukotrienes to induce tuft cells remains to be determined.

Aside from being induced by cytokines and leukotrienes, trachea and intestine tuft cells also act through their production of IL-25 and LTC$_4$ to elicit downstream responses. In the trachea, these tuft cell-produced inflammatory mediators synergize with each other to promote Type 2 inflammation, including eosinophil recruitment and ILC2 proliferation (*Ualiyeva et al., 2021*). In the small intestine, while tuft cell-produced leukotrienes are important for helminth clearance, they are dispensable for Th2 responses to protist infection (*McGinty et al., 2020*). In the lung, although IL-25 and IL-4ra signaling appears dispensable for the dysplastic epithelial response following influenza, it remains to be determined if tuft cell initiation of a Th2 response may function synergistically with leukotrienes or prostaglandins to promote chronic inflammation following severe injury.

In both the trachea and the small intestine, the sensory function of existing tuft cells serves as a starting point in a positive feedback loop triggering immune activation and tuft cell hyperplasia. In comparison, tuft cells are not present in quiescent lung. Following IAV infection, they arise from basal-like cells. The apparent increased density of tuft cells after IAV infection compared to bleomycin injury also suggest that heightened immune signaling after viral infection may play a role in promoting tuft cell formation. Based on RNA-Seq of purified tuft cells from multiple tissues (*Nadjsombati et al., 2018*), in addition to IL-25 and leukotrienes, tuft cells across many tissues also produce thymic stromal lymphopoietin, prostaglandins (*DelGiorno et al., 2020*), and acetylcholine. Most of these ligands and proteins that produce them are also expressed by damage-induced tuft cells in the lung, though we note reduced ChAT expression in post-IAV tuft cells compared to trachea tuft cells. Despite a conserved tuft cell expression signature, there are notable differences between lung injury after IAV infection and the small intestine Type 2 response, including the immune repertoires and fundamentally different epithelial progenitors which give rise to tuft cells. These differences may account for why IL-25 or IL-4/IL-13 signaling is dispensable for the basal-like cell response and tuft cell differentiation after influenza.

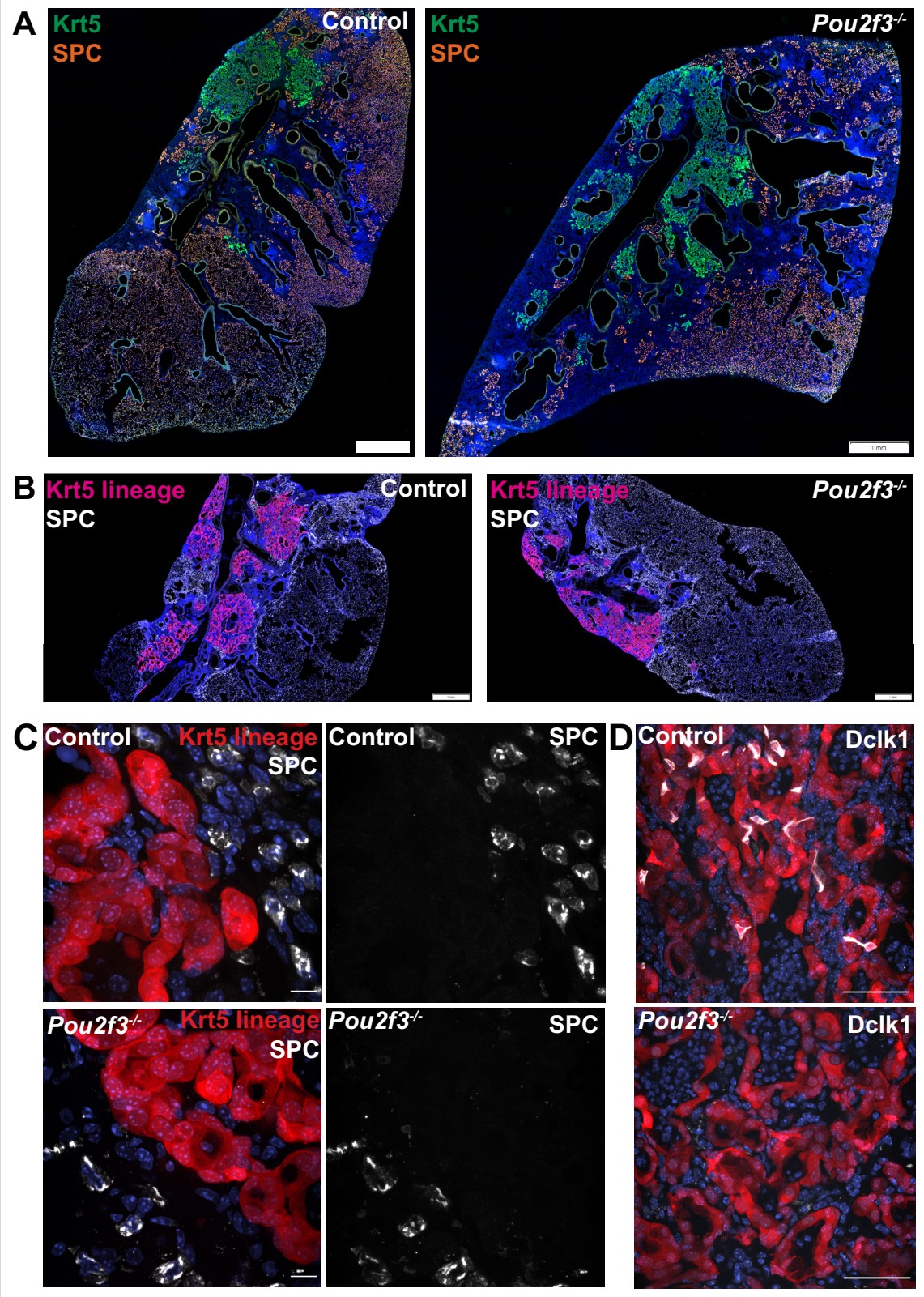

**Figure 6.** Tuft cells do not affect Krt5 plasticity following influenza. (A–B) Krt5 (green) and SPC (orange) staining in control (n=3) and *Pou2f3*[-/-] (n=3) lung sections demonstrates no appreciable overlap between Krt5 and SPC areas. (**B**) Krt5-creERT2; Ai14; *Pou2f3*[+/-] or *Pou2f3*[-/-] lung sections were injected with tamoxifen 5, 10, and 15 days post infection and lungs were harvested 30 days post infection. tdTomato signal was not found in SPC+ cells in control (n=3) or *Pou2f3*[-/-] (n=3). (A–B) Scale bar is 1 mm, (**C**) scale bar is 10 µm, (**D**) scale bar is 50 µm.

In this study, while we found that tuft cells are not required for the formation of Krt5+ cells, goblet cells, or conversion of Krt5+ cells to AT2s, we do not rule out other possible roles of tuft cells within the damaged lung epithelium. The appearance of tuft cells within the heavily injured regions is dramatic, reinforcing the need for future studies to define discrete functions of these cells in pulmonary physiology/pathophysiology. Moreover, future definition of the signals required for ectopic lung tuft cell development, apparently sufficiently distinct from the small intestine, may provide for important clues as to the enigmatic function of these intriguing cells.

## Materials and methods
### Animals and treatment
All animal procedures were approved by the Institutional Animal Care and Use Committee of the University of Pennsylvania, the University of California San Diego (UCSD), and the University of California San Francisco (UCSF): *Il25*-/- (*Fallon et al., 2006*), *Il4ra*-/- (*Noben-Trauth et al., 1997*), *Pou2f3*-/- (*Matsumoto et al., 2011*), *Trpm5*-/- (*Damak et al., 2006*), Trpm5-GFP (*Clapp et al., 2006*), *Krt5-creERT2* (*Van Keymeulen et al., 2011*), Ai14 (*Madisen et al., 2010*). *Il4ra*fl (*Herbert et al., 2004*) and *Rosa26-ERT2-Cre* (*Ventura et al., 2007*) were used for inducible knockout of *Il4ra*, animals received 2 mg tamoxifen by i.p. injection every other day from 7 to 14 days post infection. The *Ifnar1*-/- and *Il28r*-/- mice were received from the Fuchs and Striepen laboratories, respectively (University of Pennsylvania). For experiments at University of Pennsylvania, adult mice of both sexes were used in relatively equal proportions, for UCSD, 8- to 10-week-old mice (<25 g) of both sexes were used in equal proportions and all mice are on a C57BL6/J background unless otherwise noted. The protocol number associated with the ethical approval of this work is 806262 (University of Pennsylvania) and S16187 (UCSD). For all animal studies, no statistical method was used to predetermine sample size. The experiments were not randomized, and the investigators were not blinded to allocation during experiments and outcome assessment.

### IAV infection and bleomycin injury model
All viral infections utilized influenza strain A/H1N1/PR/8 obtained from Dr Carolina Lopez (*Garcia et al., 2020*). For influenza infection at the University of Pennsylvania, virus was administered intranasally. Mice ranging between 15 and 20 g in weight were infected with 30 tissue culture infectious dose (TCID)50 units of PR8, mice weighing between 20 and 25 g were given 40 TCID50 units, and mice ranging between 25 and 30 g in weight were given 50 TCID50 units. Due to increased mortality in the *Ifnar*-deficient mice (*Arimori et al., 2013*; *Seo et al., 2011*), interferon receptor-deficient mice were specifically infected with 10 TCID50 units less than their BL/6 controls. Briefly, mice were anesthetized with 3.5% isoflurane for 5 min until bradypnea was observed. Virus dissolved in 30 µl of PBS was pipetted onto the nostrils of anesthetized mice, whereupon they aspirated the fluid directly into their lungs. Infections performed at the UCSF (*Figure 3—figure supplement 3A*) were performed nearly identically, as previously described (*Vaughan et al., 2015*; *Xi et al., 2017*). A/H1N1/PR/8 infection was also performed independently at the UCSD, with virus obtained initially from ATCC (VR-95PQ). In all cases, control and experimental groups were infected simultaneously in the same cohort by the same investigator so that direct comparison between groups was justified and appropriate.

For the bleomycin injury model, mice were anesthetized as above and treated intranasally with a single dose of bleomycin (Cayman Chemicals) at 2.25 mg/kg and harvested 22 days post treatment.

### Whole-lung cell suspension preparation
Lungs were harvested from mice and single-cell suspensions were prepared as previously described (*Zhao et al., 2020*). Briefly, the lungs were thoroughly perfused with cold PBS via the left atrium to remove residual blood in the vasculature. Lung lobes were separated, collected, and digested with 15 U/ml dispase II (Thermo Fisher Scientific, #17105041) in PBS for 45 min at room temperature (RT) and mechanically dissociated by pipetting in sort buffer (DMEM +2% CC +1% P/S, referred to as 'SB'). Next, cell suspensions were filtered by the 40 µm cell strainer (Thermo Fisher Scientific, #352340) and treated by Red Blood Cell Lysis Buffer (Thermo Fisher Scientific, A1049201) for 5 min, and the cell suspension was incubated in SB containing 1:1000 DNase I (Millipore Sigma, #D4527) for 45 min at 37°C. Whole-lung cell suspensions were then used for subsequent experiments.

## Fluorescence-activated cell sorting

Whole-lung single-cell suspensions were prepared as above and then blocked in SB containing 1:50 TruStain FcX (anti-mouse CD16/32) Antibody (BioLegend, #101319) for 10 min at 37°C. The cell suspension was stained using allophycocyanin/Cy7-conjugated rat anti-mouse CD45 antibody (1:200, BioLegend, #101319), PE-conjugated rat anti-mouse EpCam antibody (1:500, BioLegend, G8.8, #118206) for 45 min at 4°C. Stained cells and 'fluorescence minus one' controls were then resuspended in SB + 1:1000 Dnase + 1:1000 Draq7 (BioLegend, #424001) as a live/dead stain. All FACS sorting was done on a BD FACSAria Fusion Sorter (BD Biosciences).

## Bulk RNA-Seq

One-thousand to 3000 EpCam + Trpm5-GFP$^+$ cells from mice at day 43 post influenza were sorted directly into lysis buffer from Takara SMART-Seq v4 and RNA/cDNA was amplified according to the manufacturer's instructions. All downstream library preparation and sequencing was performed by the Next-Generation Sequencing Core at the Perelman School of Medicine, University of Pennsylvania. In brief, libraries were sequenced on a NovaSeq sequencer at 100SR, raw FASTQ files were imported into R and reads mapped with Kallisto, and differential expression performed by Limma. Raw data is deposited at GEO.

## Single-cell RNA-Seq

Single-cell RNA-Seq was performed using the Chromium System (×10 Genomics) and the Chromium Single Cell 3' Reagent Kits v2 (×10 Genomics) at the Children's Hospital of Philadelphia Center for Applied Genomics. As with bulk RNA-Seq, ~3000 EpCam + Trpm5-GFP$^+$ cells were sorted into PBS + 0.1% BSA from mice at day 28 post influenza and loaded onto the ×10 Chromium system. After sequencing, initial data processing was be performed using Cellranger (v.3.1.0). Cellranger mkfastq was used to generate demultiplexed FASTQ files from the raw sequencing data. Next, Cellranger count was used to align sequencing reads to the mouse reference genome (GRCm38) and generate single-cell gene barcode matrices. Post-processing and secondary analysis was performed using the Seurat package (v.4.0). First, variable features across single cells in the dataset will be identified by mean expression and dispersion. Identified variable features was then be used to perform a PCA. The dimensionally reduced data was used to cluster cells and visualize using a UMAP plot. Contaminating non-tuft cells were removed by sub-setting data, requiring counts for Trpm5 > 1. Slingshot and Monocle were used for trajectory analysis. High-throughput sequence data is available at GEO.

## Tissue preparation for immunofluorescence

Each lung was thoroughly perfused with PBS via the left atrium and then perfused with 1 ml of 3.2% paraformaldehyde (PFA, Thermo Fisher Scientific) and placed in a 50 ml tube with 25 ml of PFA to shake at RT for 1 hr. Following the 1 hr incubation, the PFA was replaced with 25 ml of PBS every 20 min for the next hour while shaking at RT. The lungs were then placed in 30% sucrose (Sigma-Aldrich) overnight shaking at 4°C. The following day, the tissues were placed in 15% sucrose-50% optimal cutting temperature compound (OCT, Fisher Healthcare) shaking for 2 hr at RT. The fixed lungs were then embedded in OCT, flash frozen using ethanol and dry ice and stored at –80°C. Using a cryostat, the lungs were then section (6 µm) and stored at –20°C.

Lung sections were fixed with 3.2% PFA for 5 min at RT and then washed three times with PBS for 5 min at RT while gently shaking. Slides were then blocked for 1 hr at RT in a humid chamber with blocking buffer ([1% BSA, Gold Bio], 5% donkey serum [Sigma], 0.1% Triton X-100 [Fisher BioReagents] and 0.02% sodium azide [Sigma-Aldrich] in PBS). The slides were then stained in blocking buffer overnight at 4°C with a combination of primary antibodies. The following day, the slides were washed three times for 5 min while gently shaking at RT with PBS + 0.1% Tween (Sigma) and stained for 90 min in blocking buffer with a combination of secondary antibodies. Slides were then washed three times for 5 min while gently shaking at RT with PBS + 0.1% Tween and stained with DAPI (1:10,000 dilution; catalog no. D21490, Thermo Fisher Scientific), for 7 min and washed in PBS + 0.1% Tween as mentioned above. Slides were then mounted with Fluoroshield (Sigma) and imaged using a Leica inverted fluorescent microscope Dmi8 and analyzed using Las X and Fiji softwares.

Primary antibodies used: rabbit anti-Dclk1 (1:500 dilution; catalog no. ab37994 or ab31704, Abcam), chicken anti-Krt5 (1:1000 dilution; catalog no. 905901, BioLegend) sheep anti-eGFP (1:500,

Thermo Fisher Scientific, OSE00001G), rabbit anti-Gnb3 (1:200, 10081-1-AP, Proteintech), rabbit anti-POU2F3 (1:500, Sigma, HPA019652), rabbit anti-Agr2 (1:200, Cell Signaling Technology, 13062), rabbit anti-Muc5b (1:400, Cloud-Clone Corp, PAA684Mu01), rabbit anti-Pro-SPC (Seven Hills Bioreagents, WRAB-9337), rabbit anti-Ki67 (Abcam, ab15580). Secondary antibodies used: donkey anti-rabbit AF568 (1:1000 dilution; catalog no. A10042, Thermo Fisher Scientific), donkey anti-rabbit AF647 (1:1000 dilution; catalog no. A31573, Thermo Fisher Scientific), donkey anti-chicken AF488 (1:1000 dilution; catalog no. 703-545-155, Jackson ImmunoResearch), donkey anti-sheep AF488 (1:1000 dilution, catalog no. A11015, Thermo Fisher Scientific).

## Image quantification

For experiments performed at the UCSD (*Figures 3A–L, 5A–I and 6C–D*): lung sections were imaged on a Nikon A1 microscope using a 20×0.75 NA objective. Nikon Elements Jobs Module was used for automating multiple slide imaging, batch stitching, and maximum intensity projection of images. An analysis pipeline using Nikon Elements General Analysis 3 was used to analyze images in batch. Imaging processing split channels for individual analysis. For total tissue area, DAPI signal area was recorded. For area quantifications, a Gaussian filter was applied, an intensity threshold and size threshold was set manually, and the area of the resulting binary mask was recorded. For quantification of Dclk1$^+$ cells, a Gaussian filter was applied, and intensity and minimum size thresholds were set manually. The Dclk1$^+$ count was verified and adjusted manually. For Krt5 and SPC staining (*Figure 6A*), Krt5 lineage tracing (*Figure 6B*), and Muc5b staining (*Figure 5J–M*), left lobe lung sections were scanned on an Olympus VS200 slide scanner using a ×20 objective.

For experiments performed at the University of Pennsylvania, sections were imaged on a Leica Dmi8 as described above. Image quantification was performed by manual area measurements utilizing ImageJ/FIJI for Krt5 area and manual quantification of Dclk1$^+$ cells as at UCSD (*Figure 3M-P*) (*Figure 3—figure supplement 3A*). For interferon signaling deficient experiments (*Figure 4A-F*), the Krt5 area quantification was performed utilizing the ImageJ/FIJI software, setting the auto threshold, using the Otsu method, and manual quantifying Dclk1$^+$ cells.

## Lineage tracing

Tamoxifen dissolved in corn oil was administered by i.p. injection at 25 mg/kg.

## Lung viral titer

Lungs from *Pou2f3*$^{+/-}$ and *Pou2f3*$^{-/-}$ mice infected with PR8 were homogenized in serum-free media. Virus titer was determined by TCID50 assay using MDCK cells (ATCC, CCL34), and the presence of virus in the supernatant was determined by hemagglutination assay. Hemagglutination assay and TCID50 calculations were performed as previously published (*Xue et al., 2016*).

## Quantitative RT-PCR

Whole-lung lobes were dissected into Trizol (Invitrogen) and Rneasy Mini RNA Extraction Kit (Qiagen) was used to extract total RNA. RT-PCR was performed using iScript Select cDNA Synthesis Kit (Bio-Rad). qPCR was performed on a CFX Connect system (Bio-Rad) using SYBR Green (Bio-Rad). Three technical replicates were performed for each target gene.

| Gene name | Primer sequence |
| --- | --- |
| *Actb* | 5'-CGGCCAGGTCATCACTATTGGCAAC-3'<br>5'-GCCACAGGATTCCATACCCAAGAAG-3' |
| *Krt5* | 5'ACCTTCGAAACACCAAGCACGA-3'<br>5'-TCAGCTTCAGCAATGGCGTTCT-3' |
| *Foxa3* | 5'-CTTGGTGGAGGTTGGGTGAG-3'<br>5'-ACAGGCAGTATTCCCAAGCC-3' |
| *Agr2* | 5'-GGAGCCAAAAAGGACCCAAAG-3'<br>5'-CTGTTGCTTGTCTTGGATCTGT-3' |

*Continued on next page*

*Continued*

| Gene name | Primer sequence |
|---|---|
| *Muc5ac* | 5'-TGACTCAATCTGCGTGCCTT-3'<br>5-AGGCCTTCTTTTGGCAGGTT-3' |
| *Muc5b* | 5'-GCACGTAAATGCGACTGTCT-3'<br>5'-ATGGACCTTGCTCTCCTGAC-3' |
| *Il4* | 5'-GGTCTCAACCCCCAGCTAGT-3'<br>5'-GCCGATGATCTCTCTCAAGTGAT-3' |
| *Il4ra* | 5'-TGACCTACAAGGAACCCAGGC-3'<br>5'- GAACAGGCAAAACAACGGGAT-3' |
| *Il5* | 5'-CCTCTTCGTTGCATCAGGGT-3'<br>5'-GATCCTCCTGCGTCCATCTG-3' |
| *Il13* | 5'-AAAGCAACTGTTTCGCCACG-3'<br>5'-CCTCTCCCCAGCAAAGTCTG-3' |

## Statistics

All statistical calculations were performed using GraphPad Prism. p-Values were calculated from unpaired two-tailed t-tests with Welch's correction or ANOVA for multivariate comparisons. Variance was analyzed at the time of t-test analysis. This data is not included in the manuscript but is available upon reasonable request.

## Acknowledgements

We thank all Vaughan and Sun Lab members for helpful discussions and suggestions. We thank the CHOP Flow Cytometry Core and Center for Human Genomics, the UCSD School of Medicine Microscopy Core (NINDS P30 NS047101), and UCSD Nikon Imaging Center for assistance in performing these studies. We also thank Dr Jeffrey Gotts for performing influenza infections at UCSF. We would like to thank Dr Boris Striepen and Dr Jessica Byerly for providing us with the *Il28r$^{-/-}$* mice and Dr Serge Fuchs for providing us with the *Ifnar1$^{-/-}$* mice. Funding: This work was supported by NIH grants R01HL153539 and the Lisa Dean Moseley Family Foundation Grant to AEV, R01HL142215 to XS, 1R01AT011676 to XS, T29IR0475 to XS, NHLBI F32 HL151168 to JB, NIH F32HL140868 to MEK, T32HL007185 to MEK, AP Giannini Foundation to MEK, VA CX001617 to NAC and Postdoctoral Fellowship-Fonds de recherche du Québec-Santé to MEG. Competing interests: The authors declare that they have no competing interests. Data and materials availability: All data needed to evaluate the conclusions in the paper are present in the paper and/or the Supplementary Materials. Additional data related to this paper may be requested from the authors.

## Additional information

### Funding

| Funder | Grant reference number | Author |
|---|---|---|
| National Institutes of Health | R01HL153539 | Andrew E Vaughan |
| Lisa Dean Moseley Foundation | | Andrew E Vaughan |
| National Institutes of Health | R01HL142215 | Xin Sun |
| National Institutes of Health | 1R01AT011676 | Xin Sun |

| Funder | Grant reference number | Author |
|---|---|---|
| National Institutes of Health | T29IR0475 | Xin Sun |
| National Institutes of Health | F32HL151168 | Justinn Barr |
| National Institutes of Health | F32HL140868 | Maya E Kotas |
| National Institutes of Health | T32HL007185 | Maya E Kotas |
| A.P. Giannini Foundation | | Maya E Kotas |
| U.S. Department of Veterans Affairs | CX001617 | Noam A Cohen |
| Fonds de Recherche du Québec - Santé | | Maria Elena Gentile |

The funders had no role in study design, data collection and interpretation, or the decision to submit the work for publication.

## Author contributions

Justinn Barr, Maria Elena Gentile, Conceptualization, Data curation, Formal analysis, Investigation, Methodology, Writing - original draft, Writing - review and editing; Sunyoung Lee, Data curation, Formal analysis, Investigation, Methodology; Maya E Kotas, Data curation, Funding acquisition, Writing - review and editing; Maria Fernanda de Mello Costa, Conceptualization, Data curation, Investigation, Project administration, Writing - review and editing; Nicolas P Holcomb, Margaret McDaniel, Data curation, Formal analysis; Abigail Jaquish, Marcella Soewignjo, Data curation; Gargi Palashikar, Data curation, Investigation; Ichiro Matsumoto, Robert Margolskee, Methodology; Jakob Von Moltke, Data curation, Formal analysis, Investigation; Noam A Cohen, Funding acquisition; Xin Sun, Conceptualization, Resources, Formal analysis, Supervision, Funding acquisition, Investigation, Methodology, Writing - original draft, Project administration, Writing - review and editing; Andrew E Vaughan, Conceptualization, Resources, Data curation, Formal analysis, Supervision, Funding acquisition, Investigation, Methodology, Writing - original draft, Project administration, Writing - review and editing

## Author ORCIDs

Maria Elena Gentile ![ORCID] http://orcid.org/0000-0001-9138-1053
Xin Sun ![ORCID] http://orcid.org/0000-0001-8387-4966
Andrew E Vaughan ![ORCID] http://orcid.org/0000-0001-5740-643X

## Ethics

All animal procedures were approved by the Institutional Animal Care and Use Committee (IACUC) of the University of Pennsylvania, the University of California - San Diego, and the University of California, San Francisco. All experiments were performed with every effort to minimize suffering. The protocol number associated with the ethical approval of this work is 806262 (University of Pennsylvania) and S16187 (University of California San Diego).

## Decision letter and Author response

Decision letter https://doi.org/10.7554/eLife.78074.sa1
Author response https://doi.org/10.7554/eLife.78074.sa2

---

# Additional files

## Supplementary files
• Transparent reporting form

## Data availability
Sequencing data corresponding to Figures 1 and 2 have been deposited in GEO under accession code GSE197163. In addition to the deposited sequencing data, raw numerical data present in other figures is available in Source Data files.

The following dataset was generated:

| Author(s) | Year | Dataset title | Dataset URL | Database and Identifier |
|---|---|---|---|---|
| Vaughan AE | 2022 | Injury-induced pulmonary tuft cells are heterogenous, arise independent of key Type 2 cytokines, and are dispensable for dysplastic repair | http://www.ncbi.nlm.nih.gov/geo/query/acc.cgi?acc=GSE197163 | NCBI Gene Expression Omnibus, GSE197163 |

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
