## [Editor Report]

In this manuscript, Barr and colleagues report some novel and surprising results in regards to the development and role of tuft cells during influenza-induced lung injury. The authors demonstrate how unlike in the intestine lung tuft cells do not require Il-25, Il-4Ra, or Trmp5 but do require Pou2f3. Interestingly, loss of tuft cells in Pou2f3 null mice did not affect basal cell or goblet cell differentiation in basal cell pods, suggesting that additional studies are required to better understand the functional significance of these interesting cells.

---

## [Decision Letter]

**Decision letter after peer review:**

Thank you for submitting your article "Injury-induced pulmonary tuft cells are heterogenous, arise independent of key Type 2 cytokines, and are dispensable for dysplastic repair" for consideration by *eLife*. Your article has been reviewed by 3 peer reviewers, and the evaluation has been overseen by a Reviewing Editor and Paul Noble as the Senior Editor. The following individuals involved in the review of your submission have agreed to reveal their identity: Stijn De Langhe (Reviewer #1); Jay Rajagopal (Reviewer #2); Barry R Stripp (Reviewer #3).

Essential revisions:

These negative findings deserve to be published. Some small items that would improve the manuscript include:

1. What are the differences between normal tuft cells and these flu-induced tuft cells based on sc data.

2. What is the lineage relationship amongst the tuft 1 and tuft 2 and stressed cells computationally?

3. Clarify the tuft signaling components associated with viral infection including the type 1 and 3 interferon response genes. Can tuft cells be induced in an interferon deficient model?

4. Do Pou2f3-/- mice have any defects in viral clearance?

Although the study provides important new information, some of the findings are preliminary and can be addressed in the following:

5) What are the kinetics of tuft cell appearance following PR8 infection?

6) Are type 2 cytokines induced in the lungs of PR8 infected mice and if so, what are their kinetics of induction?

7) What, if any, changes occur to the molecular phenotype or functional properties (i.e. proliferation) of hyperplastic basal cells that appear in the lungs of PR8-infected mice?

8) The authors clearly demonstrate that hyperplastic basal cells in the lungs of PR8-infected mice are the source of rare tuft cells that appear in injured alveolar regions. In light of this, it would be helpful to track the differentiation trajectory between these cell types.

9) Tuft cell expansion following parasitic infection of the gut and associated type 2 inflammation, and basal cell differentiation into tuft cells leading to their increased abundance following lung injury, are distinct processes and likely to be regulated through distinct mechanisms. As such, the rationale for investigating the roles of type 2 cytokines in the regulation of tuft cell appearance is rather weak. In the absence of data demonstrating how basal to tuft cell differentiation is regulated, this component of the study seems preliminary.

*Reviewer #1 (Recommendations for the authors):*

I think the manuscript is well written and executed and very interesting.

I have no other questions except for what are the tuft cells really doing but I assume we will read about this in a future manuscript.

*Reviewer #2 (Recommendations for the authors):*

These negative findings deserve to be published. Some small items that would improve the manuscript include:

1. What are the differences between normal tuft cells and these flu-induced tuft cells based on sc data.

2. What is the lineage relationship amongst the tuft 1 and tuft 2 and stressed cells computationally?

3. Clarify the tuft signaling components associated with viral infection including the type 1 and 3 interferon response genes. Can tuft cells be induced in an interferon deficient model?

4. Do Pou2f3-/- mice have any defects in viral clearance?

*Reviewer #3 (Recommendations for the authors):*

The following concerns were noted:

1. The authors state that "though we anticipated a recapitulation of the circuit found in the small intestine, we observed no difference in total number of tuft cells in either IL4Ra-/- or IL25-/- animals". However, studies of tuft cells in the gut and their response to type 2 immunity, which were the basis for this line of investigation into ectopic tuft cells in the PR8-infected lung, have shown that tuft cells are part of a feed-forward loop leading to tuft cell expansion and enhanced type 2 immune responses including increased abundance of goblet cells. Since ectopic pulmonary tuft cells are derived from dysplastic basal cells after PR8 infection, rather than the reverse, this is clearly not the case in lungs of PR8 infected mice. Furthermore, since tuft cells are derived from hyperplastic basal cells in lungs of PR8-infected mice, it would seem unlikely that they impact the extent of basal cell hyperplasia.

In light of this, questions not addressed in this study include:

a) What are the kinetics of tuft cell appearance following PR8 infection?

b) Are type 2 cytokines induced in lungs of PR8 infected mice and if so, what are their kinetics of induction?

c) What, if any, changes occur to the molecular phenotype or functional properties (i.e. proliferation) of hyperplastic basal cells that appear in lungs of PR8-infected mice?

d) The authors clearly demonstrate that hyperplastic basal cells in lungs of PR8-infected mice are the source of rare tuft cells that appear in injured alveolar regions. In light of this, it would be helpful to track the differentiation trajectory between these cell types.

2. Tuft cell expansion following parasitic infection of the gut and associated type 2 inflammation, and basal cell differentiation into tuft cells leading to their increased abundance following lung injury, are distinct processes and likely to be regulated through distinct mechanisms. As such, the rationale for investigating roles for type 2 cytokines in regulation of tuft cell appearance is rather weak. In the absence of data demonstrating how basal to tuft cell differentiation is regulated, this component of the study seems preliminary.

---

## [Author Response]

Essential revisions:These negative findings deserve to be published. Some small items that would improve the manuscript include:1. What are the differences between normal tuft cells and these flu-induced tuft cells based on sc data.

We performed a direct comparison between post-flu tuft cells and previously published tracheal brush cell RNA-Seq data performed by Nadjsombati et al., (10.1016/j.immuni.2018.06.016). While there are certainly differences in gene expression, the core tuft cell signature genes are expressed at comparable levels. Interestingly some genes associated with “tuft-1” cells are expressed at a higher level in the post-flu tuft cells than in trachea brush cells. This data is now present in supplementary figure 1. In the course of this analysis we also utilized an updated RNA-Seq workflow, resulting in minor changes to the original RNA-Seq analysis, so we updated the heatmaps and volcano plots accordingly in Figure 1, and the total number of differentially expressed genes between tuft and non-tuft epithelial cells is 898.

2. What is the lineage relationship amongst the tuft 1 and tuft 2 and stressed cells computationally?

We utilized both Monocle3 and Slingshot algorithms to perform pseudotime analysis. In full agreement with our Krt5-CreERT2 lineage tracing and our previously published p63-CreERT2 lineage tracing, both algorithms predict a differentiation trajectory from the “basal -> tuft” population into the heterogenous populations of tuft cells. Both methods also predict differentiation initially into tuft-2 cells, followed by further / terminal differentiation into tuft-1 and “stressed” tuft cells (as mentioned in the original manuscript, we remain somewhat agnostic about the nature of this stressed cell population). The data are now present as supplementary figure 3.

3. Clarify the tuft signaling components associated with viral infection including the type 1 and 3 interferon response genes. Can tuft cells be induced in an interferon deficient model?

We appreciated this insightful comment and directly assessed the potential role of interferon signaling in tuft cell development. We performed influenza infections in *IL28r^-/-^* (Ifnlr1) and *Ifnar1*^-/-^ mice deficient in either type III interferon signaling or type I interferon signaling, respectively. The interferon receptor deficient mice were infected with a lower dose of PR8 than the BL/6 control mice as Ifnar deficient mice have been shown to have more severe illness when infected with flu (Arimori et al., Antiviral Research 2013, Seo et al., Plos Pathogens 2011). Even at a lower infectious dose, the interferon receptor deficient mice had an average weight loss of 22%, comparable to BL/6 controls (Supplementary Figure 9). Our data demonstrated that type I and type III interferon signaling does not play a prominent role in tuft cell abundance following flu infection (Figure 4).

4. Do Pou2f3-/- mice have any defects in viral clearance?

We investigated potential changes in viral load by comparing *Pou2f3^-/-^* and *Pou2f3^-/+^* for the presence of infectious virus at day 8 and day 12 via the hemagglutination assay. Detectable virus was variably present at day 8 in both groups, but virus was entirely cleared by day 12 in all mice (Supplementary Figure 8). Moreover, most tuft cells do not appear until later time points post-influenza. Taken together these data indicate that tuft cells do not appreciably impact viral replication or clearance.

Although the study provides important new information, some of the findings are preliminary and can be addressed in the following:5) What are the kinetics of tuft cell appearance following PR8 infection?

We apologize for not stating this more clearly in the initial submission, but we have already performed a kinetic analysis of tuft cell appearance following IAV infection in a previous manuscript. See Figure 3B in Rane et al., https://doi.org/10.1152/ajplung.00032.2019.

6) Are type 2 cytokines induced in the lungs of PR8 infected mice and if so, what are their kinetics of induction?

We performed qPCR for key Type 2 cytokines across a time course post-influenza infection, observed elevated levels of *Il4, Il5, and Il13* over the first ~10 days, but then normalized by later time points (Supplementary Figure 8A). The fact that these cytokines decrease by the time ectopic tuft cells arise in greater numbers adds additional support to our findings demonstrating that tuft cells arise independent of these Th2 signals. Our results largely agree with previous work demonstrating significant induction of *Il13* after influenza A infection (DOI: https://doi.org/10.4049/jimmunol.1800671) (DOI: 10.1038/ni.2045).

7) What, if any, changes occur to the molecular phenotype or functional properties (i.e. proliferation) of hyperplastic basal cells that appear in the lungs of PR8-infected mice?

We performed Ki67 immunostaining in Pou2f3^-/-^ and control mice, noting no difference between groups (Supplementary Figure 8C-D). This corroborates our initial analysis demonstrating no difference in total dysplastic (Krt5^+^) area between Pou2f3^-/-^ and control mice.

8) The authors clearly demonstrate that hyperplastic basal cells in the lungs of PR8-infected mice are the source of rare tuft cells that appear in injured alveolar regions. In light of this, it would be helpful to track the differentiation trajectory between these cell types.

Please see above the response to point #2. Pseudotime analysis corroborates Cre-Lox based fate mapping with both the p63-CreERT2 (Rane et al.) and Krt5-CreERT2 (present study).

9) Tuft cell expansion following parasitic infection of the gut and associated type 2 inflammation, and basal cell differentiation into tuft cells leading to their increased abundance following lung injury, are distinct processes and likely to be regulated through distinct mechanisms. As such, the rationale for investigating the roles of type 2 cytokines in the regulation of tuft cell appearance is rather weak. In the absence of data demonstrating how basal to tuft cell differentiation is regulated, this component of the study seems preliminary.

Amplification of tuft cells in the small intestine (Gerbe et al., 2016; Howitt et al., 2016; von Moltke et al., 2016) and upper airways (Ualiyeva et al., 2021, Bankova et al., 2018) are either totally dependent on or highly influenced by Type 2 cytokines, respectively. Accordingly, it was critical to examine whether a similar mechanism was at play in the lung after influenza injury, i.e. promoting tuft cell amplification downstream of Type 2 cytokines. While our findings demonstrate that post-flu tuft cells arise largely independent of Th2 signals, new findings in other tissues published after submission of the current manuscript do indeed demonstrate Th2 / ILC2-indepdent functions of tuft cells (O’Leary et al., DOI: 10.1126/sciimmunol.abj1080). Our findings support the existence of novel mechanisms regulating tuft cell differentiation, and as the Reviewer suggests, we hope to uncover these mechanisms in future work.

Reviewer #1 (Recommendations for the authors):I think the manuscript is well written and executed and very interesting.I have no other questions except for what are the tuft cells really doing but I assume we will read about this in a future manuscript.

We very much appreciate Reviewer 1’s insights, and we absolutely plan to continue these studies in the future to fully elucidate tuft cell functions in the post-influenza lung.

Reviewer #2 (Recommendations for the authors):These negative findings deserve to be published. Some small items that would improve the manuscript include:1. What are the differences between normal tuft cells and these flu-induced tuft cells based on sc data.

Please see Essential revision response #1.

2. What is the lineage relationship amongst the tuft 1 and tuft 2 and stressed cells computationally?

Please see Essential revision response #2.

3. Clarify the tuft signaling components associated with viral infection including the type 1 and 3 interferon response genes. Can tuft cells be induced in an interferon deficient model?

Please see Essential revision response #3.

4. Do Pou2f3-/- mice have any defects in viral clearance?

Please see Essential revision response #4.

Reviewer #3 (Recommendations for the authors):The following concerns were noted:1. The authors state that "though we anticipated a recapitulation of the circuit found in the small intestine, we observed no difference in total number of tuft cells in either IL4Ra-/- or IL25-/- animals". However, studies of tuft cells in the gut and their response to type 2 immunity, which were the basis for this line of investigation into ectopic tuft cells in the PR8-infected lung, have shown that tuft cells are part of a feed-forward loop leading to tuft cell expansion and enhanced type 2 immune responses including increased abundance of goblet cells. Since ectopic pulmonary tuft cells are derived from dysplastic basal cells after PR8 infection, rather than the reverse, this is clearly not the case in lungs of PR8 infected mice. Furthermore, since tuft cells are derived from hyperplastic basal cells in lungs of PR8-infected mice, it would seem unlikely that they impact the extent of basal cell hyperplasia.In light of this, questions not addressed in this study include:a) What are the kinetics of tuft cell appearance following PR8 infection?

Please see Essential revision response #5.

b) Are type 2 cytokines induced in lungs of PR8 infected mice and if so, what are their kinetics of induction?

Please see Essential revision response #6.

c) What, if any, changes occur to the molecular phenotype or functional properties (i.e. proliferation) of hyperplastic basal cells that appear in lungs of PR8-infected mice?

Please see Essential revision response #7.

d) The authors clearly demonstrate that hyperplastic basal cells in lungs of PR8-infected mice are the source of rare tuft cells that appear in injured alveolar regions. In light of this, it would be helpful to track the differentiation trajectory between these cell types.

Please see Essential revision response #2 and #8.

2. Tuft cell expansion following parasitic infection of the gut and associated type 2 inflammation, and basal cell differentiation into tuft cells leading to their increased abundance following lung injury, are distinct processes and likely to be regulated through distinct mechanisms. As such, the rationale for investigating roles for type 2 cytokines in regulation of tuft cell appearance is rather weak. In the absence of data demonstrating how basal to tuft cell differentiation is regulated, this component of the study seems preliminary.

Please see Essential revision response #9.